# Pathways to Carbon-Free Transport in Germany until 2050

**Till Gnann [1],\***　**Daniel Speth [1]**　**Michael Krail [1], Martin Wietschel [1,2] and Stella Oberle [3]**

[1]　Fraunhofer Institute for Systems and Innovation Research ISI, Breslauer Str. 48, 76139 Karlsruhe, Germany; daniel.speth@isi.fraunhofer.de (D.S.); michael.krail@isi.fraunhofer.de (M.K.); martin.wietschel@isi.fraunhofer.de (M.W.)

[2]　Institute for Industrial Production (IIP) at the Karlsruhe Institute of Technology (KIT), Hertzstrasse 16, 76187 Karlsruhe, Germany

[3]　Fraunhofer Research Institution for Energy Infrastructures and Geothermal Systems IEG, Breslauer Str. 48, 76139 Karlsruhe, Germany; stella.oberle@ieg.fraunhofer.de

\*　Correspondence: till.gnann@isi.fraunhofer.de

**Abstract:** The transport sector has to be widely decarbonized by 2050 to reach the targets of the Paris Agreement. This can be performed with different drive trains and energy carriers. This paper explored four pathways to a carbon-free transport sector in Germany in 2050 with foci on electricity, hydrogen, synthetic methane, or liquid synthetic fuels. We used a transport demand model for future vehicle use and a simulation model for the determination of alternative fuel vehicle market shares. We found a large share of electric vehicles in all scenarios, even in the scenarios with a focus on other fuels. In all scenarios, the final energy consumption decreased significantly, most strongly when the focus was on electricity and almost one-third lower in primary energy demand compared with the other scenarios. A further decrease of energy demand is possible with an even faster adoption of electric vehicles, yet fuel cost then has to be even higher or electricity prices lower.

**Keywords:** transport; emissions; alternative fuel vehicles; electric vehicles; decarbonized; energy demand; electricity; transport demand model; simulation model





## 1. Motivation

The transport sector has to largely decrease its greenhouse gas emissions to contribute to the goals set in the Paris Agreement in 2015. For this reason, a number of countries have set goals to ban fossil-fueled cars from the road within the next two decades [1]. Since passenger cars contribute most to transport emissions in industrialized countries, this is certainly the most important transport sector to address, yet heavy-duty vehicles, ships, and aviation also need attention to completely decarbonize the transport sector.

Germany, as one important industrialized country, has manifested a target to be $CO_2$ emission free by 2045 in the German Federal Climate Change Act in 2021 [2]. To reach this target, energy generation, transport, and industry have to be almost emission-free by 2045 and transport has to cut its emission by 48% until 2030. Since there was hardly any progress over the last 30 years in emission reduction from transport, this requires tremendous effort and there are numerous policy support schemes under discussion and within the coalition treaty of the current German government that could initiate such a change [3,4].

In this paper, we proposed four scenarios that could achieve a decarbonized transport sector by 2050 in Germany. These four scenarios differ in the use of energy in the modes of transport: one scenario focuses on electrification, the second on a large use of hydrogen, the third on methane produced from electricity (Power-to-gas, PtG), and the last one on liquid synthetic fuels (Power-to-Liquid, PtL), i.e., gasoline and diesel produced from electricity via electrolysis and the Fischer–Tropsch process. In all scenarios, electricity generation is decarbonized by 2050 and all electricity-based fuels will be emission free by then. PtG and PtL fuels are not locally emission free, but a production of fuels with $CO_2$ from biomass or captured from the air permits a globally emission free use. These energy carriers are used

whenever economically sensible in the scenarios. Such an analysis has been performed in other studies as well [5,6], yet here, we put special emphasis on one energy carrier per scenario to understand the demand for it. This paper is based on the results from ref. [7].

In the following Section 2, we briefly describe the methods and data used. In Section 3, results are shown and a discussion and conclusion are given in Section 4.

## 2. Methods and Data

For this analysis, we used two models that have been the basis for several publications [8–10]. The model ASTRA was used to determine the future development of transport demand in the different transport sectors. The model ALADIN was used to decide on the drive trains that will be used in future (see Figure 1 for model description and interaction).

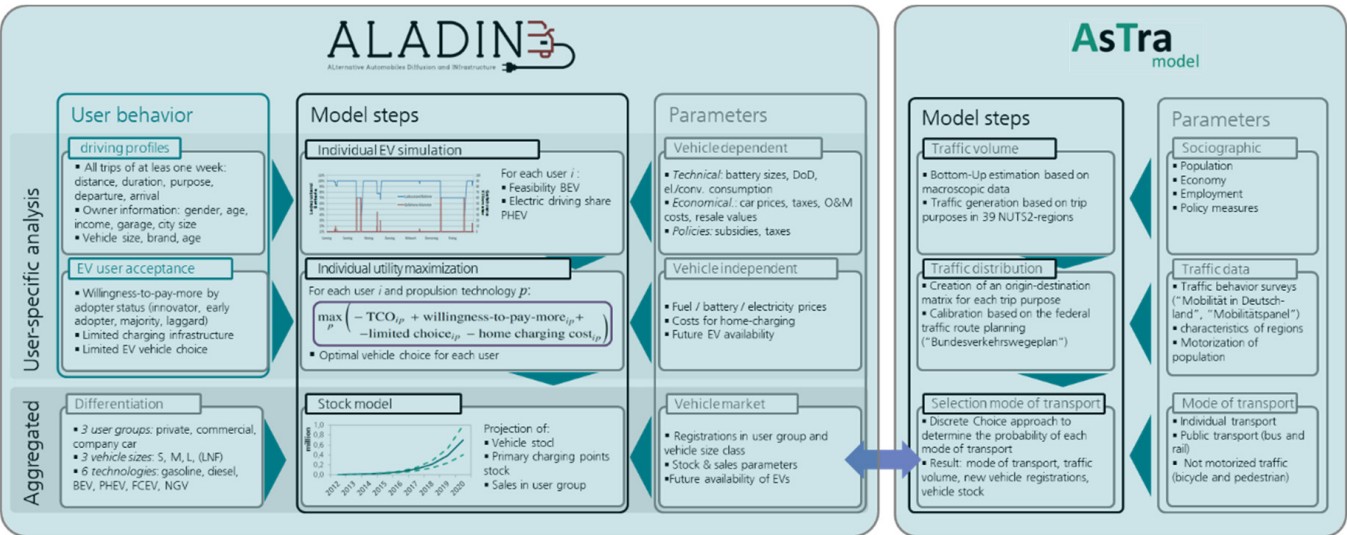

**Figure 1.** Models used and interaction for analysis.

The ASTRA model (ASsessment of TRAnsport Strategies) has been developed over more than 20 years in order to assess the impacts of transport policies on the transport system as well as on environmental indicators like GHG emissions [8]. Therefore, it assesses the future development of the national economy as well as its sectoral dynamics. On top, it includes a sophisticated population model that allows differentiating the population by attributes such as age, income, and employment status that enable an allocation of people into groups with similar mobility patterns. Based on these systems, ASTRA applies the classical 4-stage transport modeling approach (trip generation, trip distribution, modal split and assignment) to simulate the transport generation, its distribution, and the modal split both for passenger and freight transport. Therefore, it uses data generated from large mobility surveys such as the Mobility in Germany or the German Mobility Panel [11].

As ASTRA is implemented in VENSIM software based on System Dynamics methodology, it calculates the changes in the system every quarter year and allows considering feedback mechanisms such as the impact of road congestion on destination choice and modal split. Destination choice and modal split are mainly driven by changes in transport costs and transport times such that changes in drivetrain in vehicle fleets and thus increasing electrification of drivetrains induce changes in the transport related indicators like passenger- or vehicle-kilometers.

The model ALADIN (Alternative Automobiles Diffusion and Infrastructure) is a bottom-up simulation model that determines the market shares of drive trains based on individual driving profiles of vehicles. Driving largely determines the individual cost for each mode of transport. The individual total cost of ownership depending on weekly driving are calculated and accompanied by favoring and obstructing factors, e.g., for private

passenger car users. Based on these individual decisions, the market shares of the different drive trains are determined for each year and transport mode.

This very detailed analysis can only be performed where individual driving profiles are publicly available, which is the case for passenger cars and heavy-duty vehicles in Germany. All developments of market diffusion and energy demand for the other modes of transport (bus, train, inland and overseas navigation, aviation) are added based on assumptions from literature. Although international navigation and aviation are not part of the country-specific $CO_2$ budgets, these are important to receive a full picture of the transport sector. Table 1 shows an overview of modes of transport, their importance based on the final energy demand in 2019 (before the COVID-19 pandemic), and the way these transport modes are analyzed.

**Table 1.** Overview of transport modes, their final energy demand in Germany in 2019, and the modeling approach.

| Mode of Transport | Final Energy Demand in Germany in 2019 (PJ) [12] | Modeling of Future Market Diffusion |
|---|---|---|
| Passenger cars | 1549 | Individual buying decision based on TCO, favoring and hampering factors and vehicle and infrastructure availability |
| Heavy-duty vehicles | 692 | Individual buying decision based on TCO and vehicle and infrastructure availability |
| Trains | 52 | |
| Buses | 48 | |
| Inland navigation | 11 | Literature based assumptions on future development |
| Overseas navigation | 57 | |
| Aviation | 435 | |

For passenger cars, we used about 7,000 individual vehicle driving profiles derived from the German Mobility Panel [11] and the REM2030 vehicle driving database [13]. These driving profiles contain all trips of vehicles of at least one week, and additional information on the driver socio-demographics and vehicle. They were shown to be representative for German car sales in ref. [14]. For heavy-duty vehicles, the annual mileages from 6000 trucks were analyzed that stem from truckscout24 and KiD2010 [15,16] and their representativeness was shown in ref. [10].

All detailed assumptions for vehicles, framework conditions, infrastructure, and energy carrier costs were given in ref. [7]. Here, we briefly show the differing energy, battery, and fuel cell prices in the scenarios (see Tables 2–6).

The buying decision in ALADIN uses end-user prices that contain all costs for generation, transport, and distribution, but also all taxes and other levies. Further, the energy prices in the transport sector do or will in future contain a cost for $CO_2$ that has to be considered. Here, we used 200 €/t in 2030 and 500 €/t in 2050. We can observe the additional taxes and levies, e.g., in the synfuels-focused scenario with an energy carrier cost of 0.122 €/kWh in 2050 (Table 2) and a final diesel price that is 100% synthetic in 2050 (Table 3) of 0.261 €/kWh (Table 4). Energy carrier prices are varied in the scenarios for further differentiation. In focus electricity, the household electricity price is continuously reduced up to 5 €ct/kWh in 2050 because of cost savings from load shifting. In focus synfuels, we reduce fuel prices by about 2 €ct/kWh in 2050 for gasoline and diesel. This change is performed equally in focus methane with CNG and LNG prices and the hydrogen price in focus hydrogen. In focus methane, the CNG and LNG price rose until 2040 due to a large share of conventional fuels and increased afterwards because of cheaper synthetic gas prices.

Battery and fuel cell prices are the important drivers in the buying decisions of alternative drive trains. We consider common, yet optimistic prices for battery price development in focus electricity and stagnating ones in the other scenarios (cf. Table 5). The same holds

for fuel cell prices in scenarios focused on hydrogen with an optimistic development and higher price path in the other scenarios (see Table 6).

**Table 2.** Energy carrier cost for electricity-based fuels (€/kWh). All costs include transport and infrastructure cost.

| Energy Carrier Cost | 2020 | 2030 | 2040 | 2050 |
|---|---|---|---|---|
| Power-to-hydrogen | 0.285 | 0.220 | 0.150 | 0.120 |
| Power-to-gas | 0.300 | 0.195 | 0.160 | 0.122 |
| Power-to-liquid | 0.300 | 0.205 | 0.170 | 0.132 |

**Table 3.** Assumed share of synthetic fuels in transport.

| Scenario | 2020 | 2030 | 2040 | 2050 |
|---|---|---|---|---|
| Focus—electricity | 0% | 10% | 20% | 50% |
| Focus—hydrogen | 0% | 10% | 20% | 50% |
| Focus—methane | 0% | 20% | 50% | 100% |
| Focus—synfuels | 0% | 20% | 50% | 100% |

**Table 4.** Energy carrier prices (€/kWh).

| Energy Carrier Price | 2020 | 2030 | 2040 | 2050 |
|---|---|---|---|---|
| Gasoline price | 0.154 | 0.233 | 0.315 | 0.293 |
| Diesel prices [1] | 0.117 | 0.197 | 0.281 | 0.261 |
| Hydrigen price [2] | 0.285 | 0.285 | 0.282 | 0.235 |
| CNG price [3] | 0.088 | 0.190 | 0.273 | 0.257 |
| LNG price [3] | 0.097 | 0.212 | 0.317 | 0.304 |
| Electricity price households [4] | 0.329 | 0.321 | 0.313 | 0.311 |
| Electricity price commercial [4] | 0.226 | 0.217 | 0.210 | 0.208 |
| Electricity price industrial [4] | 0.130 | 0.131 | 0.136 | 0.135 |

[1] Lower in the synfuels-focused scenario, [2] Lower in the hydrogen-focused scenario, [3] Lower in the methane-focused scenario, [4] Lower in the electricity-focused scenario.

**Table 5.** Battery prices (€/kWh). Own assumptions based on refs. [17,18].

| Scenario | EV Type | 2020 | 2030 | 2040 | 2050 |
|---|---|---|---|---|---|
| Focus—electricity | BEV | 240 | 100 | 90 | 80 |
| | PHEV | 264 | 110 | 98 | 88 |
| Focus—hydrogen | BEV | 240 | 100 | 100 | 100 |
| | PHEV | 264 | 120 | 120 | 120 |
| Focus—methane | BEV | 240 | 120 | 120 | 120 |
| | PHEV | 264 | 132 | 132 | 132 |
| Focus—synfuels | BEV | 240 | 120 | 120 | 120 |
| | PHEV | 264 | 132 | 132 | 132 |

**Table 6.** Fuel cell prices (€/kW). Own assumptions based on refs. [17–20].

| Scenario | 2020 | 2030 | 2040 | 2050 |
|---|---|---|---|---|
| Focus—electricity | 234 | 80 | 66 | 55 |
| Focus—hydrogen | 234 | 78 | 62 | 50 |
| Focus—methane | 234 | 80 | 80 | 80 |
| Focus—synfuels | 234 | 80 | 80 | 80 |

## 3. Results

Let us now turn to results. Figure 2 shows the vehicle stock of passenger cars, light-duty trucks, medium-duty trucks, and heavy-duty trucks in rows 1–4 and the four scenarios in columns 1–4.

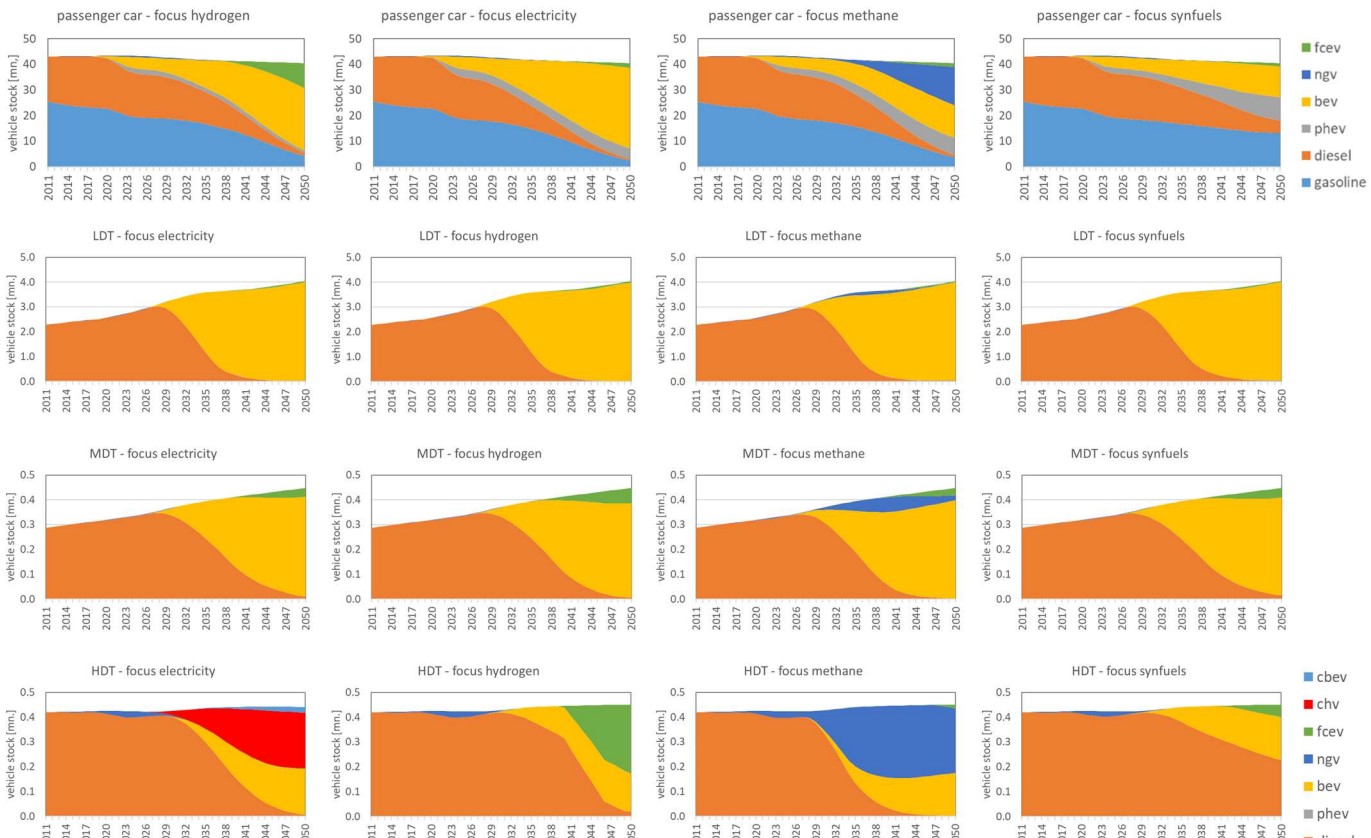

**Figure 2.** Vehicle stock in the four scenarios (**columns**) and vehicle size classes (**rows**). phev: plug-in hybrid electric vehicles: bev: battery electric vehicles; ngv: natural gas vehicles; fcev: fuel cell electric vehicles; chv: catenary hybrid electric vehicles; cbev: catenary battery electric vehicles.

For passenger cars, we find at least 25% and up to 70% BEV in 2050. The number of gasoline and diesel cars are up to 30% in the first three scenarios and some reach 50% in focus synfuels. Hydrogen and NGV only play a role if their fuel prices are very low. Thus, a quite large number of passenger cars will be electric vehicles while the rest depends on fuel prices for synthetic fuels and hydrogen.

It is less complicated for light-duty vehicles (up to 3.5 t), which will become electric in all scenarios. Medium-duty trucks (3.5–12 t) all contain a large number of battery electric trucks in 2050 in all scenarios (at least 50%), while all other transport modes depend on energy prices. This is similar for heavy-duty trucks where 30% are battery electric trucks and the other share are similar to medium-duty trucks. In focus electricity, the leftover vehicles are catenary hybrid vehicles; focus hydrogen, methane, and synfuels cover their long-distance trips with trucks powered by these fuels.

Thus, quite a large number of road transports will be powered directly with electricity while longer distances are covered with drive trains of the focus scenario.

Figure 3 contains exemplary results for the final energy demand in the transport sector in all four scenarios in 2030 and 2050 distinguished by energy carrier. In 2030, we find only some energy demand of alternative drive trains mainly stemming from the electricity demand of passenger cars. Heavy-duty vehicles can partly be fueled alternatively with electricity, hydrogen, or methane. There is a small energy demand for methane-powered vehicles in the methane-focused scenario.

In 2050, these differences are growing further. In focus electricity, earthbound transport and national aviation is carried out electrically, while the other sectors rely on liquid fuels due to technical restrictions. When counting on liquid fuels (focus PtL), the heavy-duty sector, national aviation, and parts of the railway system use synthetics or biofuels, resulting in a higher annual final energy demand. Hydrogen could play a role in the same transport

sectors and partly also for long-distance driving passenger cars as could methane. The total final energy demand would be higher due to the lower efficiency of these drive trains.

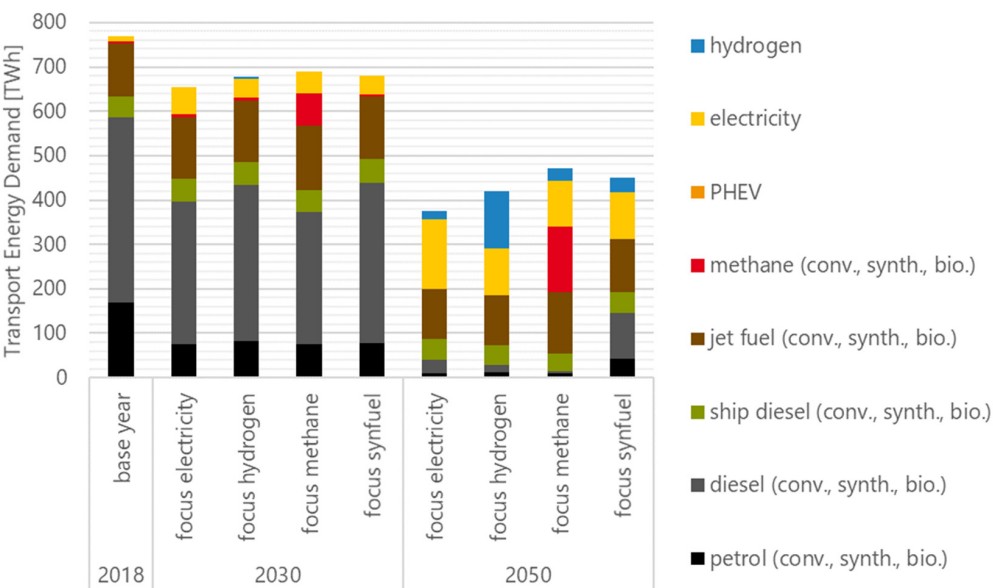

**Figure 3.** Final energy demand in 2030 and 2050 distinguished by energy carrier in the four scenarios.

Assuming that no biofuels are available for transport since they are needed in other sectors for a complete decarbonization, all fuels would have to be produced from electricity by 2050. If this was the case, the electricity demand for transport would be as shown in Figure 4. We can clearly observe that a large electrification would cause the lowest electricity demand for transport (~700 TWh) compared with the synfuels-focused or methane-focused scenario (~1000 TWh). However, the large shares of liquid or gaseous fuels in all scenarios showed the increasing importance of the aviation and navigation sectors that are not expected to be largely electrified by 2050.

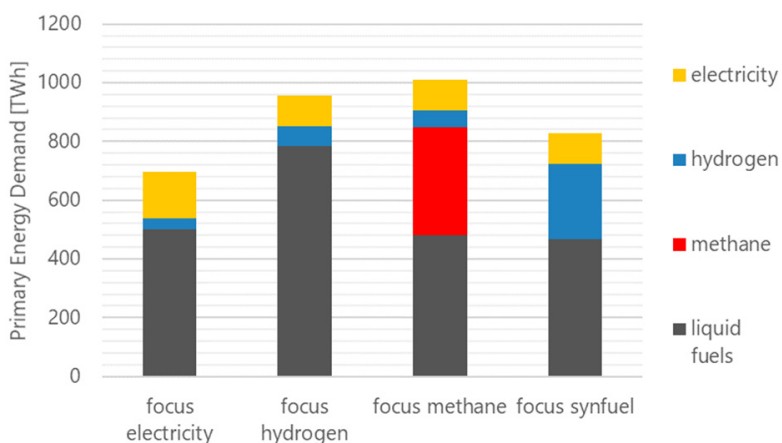

**Figure 4.** Electricity demand in 2050 distinguished by energy carrier type in the four scenarios.

To further bring down the share of conventional vehicles, we calculated sensitivities of energy prices on conventional vehicle market shares in 2050 in the elecricity-focused scenario. These are shown for diesel and gasoline vehicles for passenger cars in the first row and for light-, medium-, and heavy-duty vehicles in the second row of Figure 5.

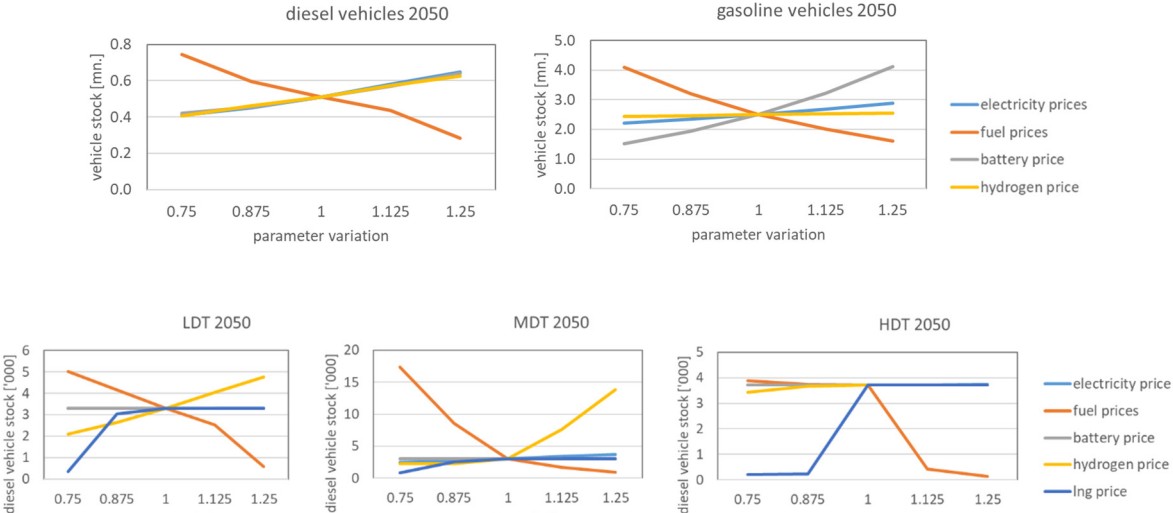

**Figure 5.** Sensitivities on conventional vehicle stock in 2050 in the electricity-focused scenario for changes in energy prices. Variation of energy prices by −25% to +25% shown on the x-axis and respective results for diesel or gasoline passenger cars (upper panel) and diesel vehicles in LDT, MDT, and HDT (lower panel). Changes for electricity price (blue), fuel (gasoline + diesel) prices (orange), battery price (gray), hydrogen price (yellow), and LNG price for trucks (dark blue).

We find the potential to further reduce gasoline and diesel passenger cars by almost half if we reduce electricity prices or increase synfuel prices by 25%. With the same changes, we could decrease the diesel truck stock in all size classes to almost zero in 2050. Thus, these changes can be considered in an even more ambitious scenario towards greenhouse gas neutrality in 2045.

## 4. Discussion and Conclusions

This paper described pathways for a complete decarbonization of transport in 2050 in Germany. These pathways are based on simulations with the models ASTRA and ALADIN and subject to a number of assumptions. Although the models have been used in multiple earlier publications, it is still discussible whether the approach is appropriate. Yet, based on literature reviews for market diffusion models in the transport sector, it is rather common [21–23]. Furthermore, the assumptions for all input parameters are subject to debate. All assumptions are based on literature and we performed a sensitivity analysis for the impact on model results for energy carrier prices to cope for this aspect.

What can we conclude from this analysis? We found high shares of electric vehicles in all scenarios and this will be the most energy efficient and economical option in the future. When batteries cause range limitations due to a lower energy density, the most cost effective energy carriers with higher ranges will be considered. In the scenarios, there can be some hydrogen vehicles if the hydrogen price is very low or also methane (or synfuels)-powered vehicles have a low price for methane (or synfuels). Yet, this comes at a price related to primary energy necessary for their production. We need about 300 TWh of additional electricity from renewables if converted to methane or synfuels and used in the transport sector. For comparison, the total renewable electricity generation in Germany was ~250 TWh in 2020, which shows the need for an efficient use of energy and direct electrification where possible. This will raise the question of importing these energy sources to Germany. Such import options do not exist at present and must first be built up with enormous investments. Furthermore, against the backdrop of the Russian war against Ukraine, questions of energy security then arise.

In any case, it is necessary to mention that these scenarios are very ambitious in assumptions and outcome and that the decarbonization of the transport sector will require large efforts.

**Author Contributions:** Conceptualization, T.G. and D.S.; methodology, T.G., D.S., M.K. and M.W.; software, T.G., D.S. and M.K.; validation, T.G., D.S. and M.K.; formal analysis, T.G., D.S. and M.K.; investigation, T.G., D.S., M.K., M.W. and S.O.; resources, M.W. and S.O.; data curation, T.G., D.S. and M.K.; writing—original draft preparation, T.G. and D.S.; writing—review and editing, T.G., D.S., M.K., M.W. and S.O.; visualization, T.G. and D.S.; supervision, M.W.; project administration, M.W. and S.O.; funding acquisition, M.W. All authors have read and agreed to the published version of the manuscript.

**Funding:** This research was funded by the Federal Ministry for Economic Affairs and Climate Protection (BMWi) in the projects "Long-term scenarios for the transformation of the energy system in Germany 3" and "MethQuest_MethSys-Production and use of methane from renewable sources in mobile and stationary applications; subproject: Energy system analysis for the production and use of methane from renewable sources" (FKZ 03EIV046A).

**Institutional Review Board Statement:** Not applicable.

**Informed Consent Statement:** Not applicable.

**Conflicts of Interest:** The authors declare no conflict of interest.

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
