# Peer review of "Pathways to Carbon-Free Transport in Germany until 2050"

_wevj, doi:10.3390/wevj13080136_

Round 1

Reviewer 1 Report

The manuscript reported the four pathways to a carbon-free transport sector in Germany in 2050, including electricity, hydrogen, synthetic methane, or synthetic liquid fuels. A transport demand model is adopted for future vehicle use and a simulation model is used for the determination of alternative fuel vehicle market shares. It is found that a further decrease in energy demand is possible with even faster adoption of electric vehicles, yet fuel cost has to be even higher or electricity prices lower.

I consider the content of this manuscript will definitely meet the reading interests of the readers of the World Electric Vehicle Journal journal. However, there are certain English spelling and grammar issues, and also the discussion and explanation should be further improved.

Therefore, I suggest giving a minor revision and the authors need to clarify some issues or supply some more experimental data to enrich the content. This could be comprehensive and meaningful work after revision.

See the PDF file for more details.

Reviewer 2 Report

- In order to fully understand the paper it is necessary to read source 4.

- Sentence in line 66-68 "Driving ..." incomprehensible

- Table description ABOVE table

- Table 3: still 50% synthetic fuels by 2050 on focus electricity and hydrogen?

- Table 4: CNG and LNG prices are getting more expensive, but in Table 2  PtG and PtL are getting cheaper?

- Figure 2:

- Please in landscape format, as it is very small.

- Retain the order of the scenarios (also in the following illustrations), otherwise it would be confusing.

- Harmonize axis labels! Y axis is confusing in 3rd and 4th row.

- Use same colors throughout the legend, it is very difficult to understand otherwise. Abbreviations are not mentioned in the text, e.g. what is "obev"?

- Figure 3 shown twice.

- Figure 5: The description is not enough to understand the figure.
